# A Bioinformatic Assay of Quercetin in Gastric Cancer

**DOI:** 10.3390/ijms25147934

**Published:** 2024-07-20

**Authors:** Sergio Raúl Zúñiga-Hernández, Trinidad García-Iglesias, Monserrat Macías-Carballo, Alejandro Pérez-Larios, Yanet Karina Gutiérrez-Mercado, Gabriela Camargo-Hernández, Christian Martín Rodríguez-Razón

**Affiliations:** 1Departamento de Ciencias de la Salud, Centro Universitario de los Altos, Universidad de Guadalajara, Tepatitlán de Morelos 47620, Mexico; 2Instituto de Investigación de Cáncer en la Infancia y Adolescencia, Departamento de Fisiología, Centro Universitario de Ciencias de la Salud, Guadalajara 44340, Mexico; trinidad.giglesias@academicos.udg.mx; 3Laboratorio de Biociencias, Departamento de Clínicas, Centro Universitario de los Altos, Tepatitlán de Morelos 47620, Mexico; monserrat.macias@cualtos.udg.mx; 4Laboratorio de Nanomateriales, Agua y Energia, Departamento de Ingenierias, Centro Universitario de los Altos, Tepatitlán de Morelos 47620, Mexico; alarios@cualtos.udg.mx; 5Laboratorio Biotecnológico de Investigación y Diagnóstico, Departamento de Clínicas, Centro Universitario de los Altos, Tepatitlán de Morelos 47620, Mexico; yanet.gutierrez@academicos.udg.mx; 6Instituto de Investigación en Ciencias Médicas, Centro Universitario de los Altos, Universidad de Guadalajara, Tepatitlán de Morelos 47620, Mexico; gabriela.camargo@academicos.udg.mx; 7Laboratorio de Experimentación Animal (Bioterio), Departamento de Ciencias de la Salud, Centro Universitario de los Altos, Tepatitlán de Morelos 47620, Mexico

**Keywords:** gastric cancer, quercetin, bioinformatics

## Abstract

Gastric cancer (GC) remains a significant global health challenge, with high mortality rates, especially in developing countries. Current treatments are invasive and have considerable risks, necessitating the exploration of safer alternatives. Quercetin (QRC), a flavonoid present in various plants and foods, has demonstrated multiple health benefits, including anticancer properties. This study investigated the therapeutic potential of QRC in the treatment of GC. We utilized advanced molecular techniques to assess the impact of QRC on GC cells, examining its effects on cellular pathways and gene expression. Our findings indicate that QRC significantly inhibits GC cell proliferation and induces apoptosis, suggesting its potential as a safer therapeutic option for GC treatment. Further research is required to validate these results and explore the clinical applications of QRC in cancer therapy.

## 1. Introduction

Gastric cancer (GC) is one of the most common types of cancer worldwide [1], with the worst prognosis in developing nations, especially those struggling with food safety since they have a higher incidence of some factors associated with the apparition of GC such as infection by *Helicobacter pylori* [2].

According to the Global Cancer Observatory (GLOBOCAN) data from 2022, GC has a mortality rate of up to 30% depending on the country [3]. Furthermore, once diagnosed, the treatment varies, going from more “traditional” radiotherapy or chemotherapy to surgery. However, those treatments are highly invasive and, since GC involves key organs needed for obtaining nutrients, these treatments can also be dangerous for the patient [4]. Therefore, there is an actual need for a safer, cheaper, and less aggressive treatment and one alternative could be treatment originating directly from botanical products, such as whole plants, or biocompounds originating from them.

Quercetin (QRC) is a flavonoid compound found in many products, including onions, cherries, broccoli, fruits, and plants like *Hibiscus sabdariffa* [5]. It has been linked to multiple beneficial effects, such as acting as a hypotensive, antioxidant, hypoglycemic, or anticarcinogenic agent [5,6,7,8,9,10,11,12]. Those effects have been attached to many different mechanisms in different kinds of cancer; for instance, in breast cancer cells (T47D), QRC interrupts the cell cycle by arresting it in the G2/M phase [6]. Furthermore, other QRC mechanisms against cancer have been detected like changes in the BCL anti-apoptotic proteins, increases in caspase releases, etc. [12]. In a previous study, potential therapeutic targets of QRC in human cells were tested, and one result was the implication of QRC in GC [13]. However, this result was only an implication and lacked further support.

Therefore, the objective of this study was to obtain a more concise picture of the effects of QRC in GC through a network pharmacological analysis using a plethora of computer databases. This strategy identified potential therapeutic targets of QRC in GC, which were then subjected to gene ontology tests, a protein-to-protein analysis, survival tests, an immune system infiltration analysis, and a molecular docking analysis. Finally, the results were compared with current information to corroborate them with GC or cancer in general.

## 2. Results

### 2.1. Target Genes

#### Common Genes between GC and QRC

Figure 1 shows a Venn diagram from Venny 2.1 (https://bioinfogp.cnb.csic.es/tools/venny/ (accessed 13 November 2023) comparing QRC- and GC-related genes according to Swiss Target Prediction (http://www.swisstargetprediction.ch/) (accessed 13 November 2023) and Malacards (https://www.malacards.org/) (accessed 13 November 2023), respectively. There are 31 genes in common.

### 2.2. Gene Ontology Test (GO)

The results for the gene ontology assay are shown in Figure 2, Figure 3, Figure 4 and Figure 5 using the information obtained from the ShinnyGo 0.8 website (http://bioinformatics.sdstate.edu/go/) (accessed 13 November 2023); these include the Pathway Assessment Chart, the Pathway Network Analysis, and the KEGG pathway, with the highlighted genes directly related to gastric cancer. Figure 5 shows the cellular component analysis using the DAVID Bioinformatics database (https://david.ncifcrf.gov/tools.jsp)(accessed 13 November 2023).

### 2.3. Protein-to-Protein Analysis 

The protein-to-protein (PPI) analysis from StringDB (https://string-db.org/) is shown in Figure 6, containing relationships between proteins coded from the 31 common genes. The 30 genes with the most interactions were chosen as HUB genes for the survival and infiltration analyses.

### 2.4. Survival Graphs

Figure 7 displays the survival graph for AXL, the only gene that showed statistical significance, along with others that were closer to reaching statistical significance for reference. 

### 2.5. Immune System Infiltration

Figure 8 shows the immune infiltration for each gene, as determined by the TISIDB website (http://cis.hku.hk/TISIDB/). This section only displays information for cells with the closest rho value to −1 and/or 1.

### 2.6. Docking Analysis

For the molecular docking analysis, Cytoscape was used to determine which HUB genes are related to mechanisms activating cancer invasion and metastasis. These were MET, EGFR, MMP2, MMP9, TOP2A, AKT1, KDR, SRC, PTK2, IGF1R, PIK3CG, PIK3R1, and GSK3B. The SwissDock website (http://www.swissdock.ch/) (accessed 13 November 2023) was then used to perform a molecular docking analysis between the proteins of the final 13 genes and QRC, and the results were visualized using UCSF CHIMERA software Version 1.17.3 (https://www.cgl.ucsf.edu/chimera/docs/ContributedSoftware/webdata/webdata.html#browser (accessed 4 April 2024). The docking figures are shown in Figure 9.

## 3. Discussion

Bioinformatic tools have emerged in the last two decades as a collection of strategies that help determine potential therapeutic targets of various chemical compounds, making further research more precise and with a higher degree of certainty. The results of this study suggested a relationship between the QRC and GC pathways [13], leading to the conception and presentation of a more concise and complete analysis of such relationship. 

The results for the gathering of genes are presented in Figure 1; we compared the genes associated with GC and the potential therapeutic targets of QRC using the Venny 2.1.0 website. The information for each gene was obtained from two different websites. As for the GC-associated genes, the database was Malacards [14], which provides extensive information and has been used in several publications [15,16]. Here, the information on all the GC genes was obtained using an algorithm that generates a comprehensive list by taking into account the known GeneCards Search mechanism, genetic testing resources, genetic variation resources, and the manually curated association of a disease with specific genes. On the other hand, QRC target genes were obtained through Swiss Target Prediction. This tool analyzes the molecule of interest in five dimensions using the Tanimoto Index [13,17]. These two tools have been widely used in studies aimed at identifying molecular targets broadly or in specific diseases [18]. As such, using these tools to identify relevant genes for this study was crucial, as shown in Figure 2.

The gene ontology assay results are displayed in Figure 3; here, the associated pathways of the 31 genes common to QRC and GC were tested. Among these results, several are immediately interesting. For example, in stem cancer cells, EGFR (Epidermal growth factor receptor) is a key tyrosine kinase that regulates the initiation, maintenance, and survival of cancer cells [19]; it has also been reported that mutations in this gene trigger the onset of other types of cancer [20], and it is known that in GC, EGFR acts as an oncogene [21]. In general, EGFR pathways are crucial for understanding cancer mechanisms. Other interesting pathways include the PI3K-Akt signaling pathways, proteoglycans in cancer, and focal adhesion, as both proteoglycans and focal adhesion play key roles in cancer propagation and potential metastasis [22]. Another pathway of interest is the chemokine signaling pathway, not only because cancer cells are known to produce pro-inflammatory molecules [23] but also because chemokines are important in immune system regulation and infiltration, which are processes linked to cancer propagation or mitigation [24], including (but not limited to) GC. All the signaling pathways obtained in the GO analysis were also tested in a network analysis (Figure 3), showing that most are closely linked to one another, suggesting that changes in one system could produce changes in all the other linked pathways. Furthermore, in Figure 4, the KEGG pathway of GC is shown with the key genes identified in the comparison of QRC and GC highlighted in red. The information presented by this database is similar to the GO analysis as most of the highlighted molecules are linked to the EGFR, PI3K-Akt, inflammation, and cell cycle processes of GC. These results are not only indicated by bioinformatic tools but are also supported by some publications that prove the importance of the suggested signaling pathways from the GO analysis [25]. Finally, regarding gene ontology, Figure 5 shows an analysis using the DAVID Bioinformatics database, giving insight into the cell structures linked to the 31 genes of interest. The most interesting results are the annotations for the nucleus and the cytoplasm of the cells. These structures are crucial in many cell processes, but particularly in cell cycle regulation and cell replication for any type of cancer. These results are not surprising, as studies have suggested how QRC can modulate the cell cycle in cancer cells. For instance, Chou et al. found that QRC can arrest the cell cycle and activate apoptosis in human breast cancer MCF-7 cells [26]. It is also important to note that while the pathways suggested to be modified by QRC in GC are presented here, the relationship between the genes in any disease is not only linked to their pathways but also by how the proteins transcribed from these genes interact with each other, as shown by the PPI network analysis.

The PPI network analysis and obtention of the HUB genes are shown in Figure 6. Here, there is an interconnected network between most proteins derived from the genes of interest. This result aligns with expectations, as historically many publications corroborate the idea that protein interactions are closely linked to cancer [27]. This level of interconnection has many layers, as StringDB proclaims. The color of the strings indicates whether interactions are directly curated from databases (light blue), experimentally determined (purple), predicted by in silico analysis (green, red, and navy blue), or another kind of interaction analysis: text-mining (light green), co-expression (black), and protein homology (gray). As briefly mentioned, most proteins have a higher degree of interconnection in cancer. However, most relationships between each HUB gene (and their corresponding proteins) have not been fully studied. Even so, some cancer studies have looked to add such interactions. Kim et al. [28], in a study comparing the effectiveness of two ginseng variants in SVEC4-10 cells to generate an anti-angiogenic effect, found that ginseng was capable of this, with the proposed mechanism linked to a phosphorylation chain of FAK, Src, Akt, and ERK diminishing VEGF-R2. This result is of interest for this study as it directly links at least two HUB genes (Src and Akt).

Furthermore, Sigstedt et al. [29] found a similar effect when testing a *Taraxacum officinale aqueous* extract. Still, they also found that the phosphorylation of genes like FAK reduced metalloproteases (MMP2 and MMP9), decreasing breast and prostate cancer cell invasion by promoting PIK-Akt pathway activity and keeping Nf-kB in the cytoplasm of the cancer cell. This result is not isolated; Villegas-Comonfort et al. tested arachidonic acid’s capacity to promote migration and invasion in MDA-MB-231 breast cancer cells [30] and found a close relationship between PI3K-Akt and both migration and invasion, also by promoting Nf-kB activity. All these results are noteworthy since most biocompounds tested against cancer revealed some interaction to highlight. However, one interaction not presented is the impact of HUB genes on other aspects of cancer as a disease, for example, survival.

Using the HUB genes obtained through the PPI network analysis (30 genes total), a survival analysis was performed using GEPIA2. Survival curves have been useful in determining the efficiency of certain clinical treatments or the progression of cancer types [31]. Nowadays, they are also useful to see the importance of any protein/gene in survival odds against any cancer. As for the results presented in Figure 7, only one HUB gene, AXL, had statistical significance in this test. This gene is transcribed and translated to the AXL receptor tyrosine kinase. AXL has been identified as a receptor that transduces signals from the extracellular matrix into GAS6, regulating biological processes like cell survival, proliferation, migration, and differentiation [32]. One proposed mechanism for AXL to enable these processes is through transcriptional regulation of MMP9 (another HUB gene in this study) [33]; Tai et al. [34] showed how, when AXL transcription increases, several cancer cell lines activate the Nf-κB pathway, concluding with an increase in MMP9 levels, a protein closely related to metastasis. Nevertheless, to our knowledge, the relationship between AXL and GC has not been fully tested. Interestingly, Nf-κB is a common mechanism between numerous processes related to cancer evolution, including metastasis and immune system infiltration, a process discussed next.

Since their seminar study delimited the important factors for cancer appearance and evolution, Hanahan and Weinberg [35] made it clear that the immune system is a very complicated but important part of this disease. For instance, there is a relationship with the type of macrophage cells surrounding tumor cells. If the macrophages are of the M2 kind, they will promote cancer proliferation (by promoting angiogenesis) [36]; thus, M2 cells promote the ideal microenvironment for cancer. On the other hand, M1 macrophages can do the opposite; M1 cells are adept at combating the disease. Just like this example, many cells of the immune system can have either of those roles. In this study, the correlation between the HUB genes and GC was tested. Figure 8 is divided into parts. In part (a), the only results shown are those of the CD4+ type cells. CD4+ cells have been associated with a predominant role in the progression and evolution of infectious and autoimmune diseases [37]. However, in the last decades, the role of CD4+ cells in cancer prognosis and treatment has become more interesting. For instance, Zhen-Quan et al. showed how CD39 expression of CD4+ cells is associated with poor survival in GC [38]. Also, since our results suggest that most of the CD4+ cells are not statistically significant with some of the HUB genes (SYK, PARP1, CA9, PI3KR1, SRC, ABCGW, PLK1, TERT, CDK2, EGFR, MMP9, ABCC1, GSK3B) but all of them have a rho value other than 0, the idea that those genes have a key role in the infiltration for this kind of cell is reinforced. However, the implication of the presence of any gene and CD4+ cells is not something that this methodology can assess, but it is possible to make some connections when looking at other research. Similar results can be observed in other kinds of immune cells as well. CD8+ cells showed how, for most cases, there is a negative correlation (rho equal to or less than 0) between the HUB genes and the infiltration of this kind of cell in GC. This could be because the roles of some of the tested genes like AXL or PI3CG are linked to the activation of metabolic pathways that increase the energy consumption of the cancer cells [39]. Also, there is an important relationship between the energy consumption of CD8+ cells and their proliferation and survival in cancer [40]. The malignant cells will outcompete the immune cells and, as such, cause them to decrease in number in the disease. Furthermore, most cancers trigger “sickness-induced anorexia”, which reduces the number of nutrients available for all cells and, therefore, can reduce the infiltration of CD8+ cells to tumors [41]. Our results could be important since, for example, QRC can, in the case of GC, reduce the transcription of the genes that increase tumor energy consumption. This would be a mechanism in which QRC helps in cancer treatment, especially in GC, which is a type of cancer that by itself makes it harder for the patient to obtain nutrients.

Another important result from Figure 8 is shown in part (c), where the infiltration of other cells of the immune system in GC was tested. For the mast cells, the result of the analysis is mixed, with the HUB genes having both a positive and negative correlation. This result is significant due to the relationship of mast cells with cancer. At the early stages of the appearance of malignant cells, mast cells are key in the recruitment of the immune system for their destruction [42]. However, mast cells can increase the angiogenesis of tumors and promote the infiltration of M2 macrophages (and other cells) that promote metastasis and cancer progression [36]. As such, further understanding of QRC and its relationship to the activation of mast cells is important but outside the scope of this study. Finally, the last cells to be discussed are eosinophils, neutrophils, and NK cells. While most of their infiltration in GC is positively correlated to the HUB genes, their role in the disease closely mirrors that of the mast cells [42,43], but the understanding of their role in GC and the effects of QRC is still limited. The biggest limiting factor of this test is that, while we can know if the HUB genes that are affected in GC by QRC have an impact on immune system infiltration, it cannot be known if the interaction from QRC will increase or reduce the transcription of a certain gene, or even if the interaction between the molecules can happen to begin with. Fortunately, while the levels of protein transcriptions are beyond the scope of this study, the molecular docking is not.

Finally, Figure 9 shows the docking analysis results of the genes linked to metastasis in GC (filtered by Cytoscape). This analysis was performed using the SwissDock website, which uses EADock DSS. This algorithm uses a binding model within every potential 3D cavity to identify the targets of chemicals on proteins. Grosdidier A., Zoete V., and Michielin O. report that this software has a near-70% accuracy rate in properly predicting binding models in this task [44]. Additionally, it uses tools like Chemistry at Harvard Macromolecular Mechanics (CHARMM) and Fast Analytical Continuum Treatment of Solvation (FACTS) [45], which help it distinguish and filter its results. In combination, these tools allow ligands with less than 15 free dihedral angles and/or test complexes with defined binding pockets to achieve a 96% success rate in EADock DSS. Later, to analyze the obtained data, attention was focused on the “Delta G” of each of the proteins with QRC. The lower the Delta G value, the higher the molecule’s bond to the protein. According to this, most of the proteins related to the metastasis of GC can be bound to QRC; out of the 13 genes tested in this analysis, 10 had a ΔG ≤ −6, so they were shown. This result supports the idea that QRC can have an effect on these gene-related pathways, and that effect could be started by the chemical binding of QRC to their proteins.

## 4. Materials and Methods

### 4.1. Bioinformatic Analysis

Figure 10 shows a diagram containing the general workflow of the entire study.

#### 4.1.1. Gathering of Target Genes

Quercetin (QRC) and gastric cancer (GC) were chosen as the key concepts of the study. They were subjected to a bioinformatic analysis individually and in conjunction, as described below.

##### QRC Target Genes

The website Swiss Target Prediction [17] was used to gather the key targets of quercetin in human cells. This required the SMILE sequence of QRC, obtained from the PubChem website, which was later used for the Swiss Target Prediction analysis.

##### GC Target Genes

The genes related to GC were gathered from the Malacards website [14] by searching for the term “Gastric Cancer” and choosing the corresponding option with MCID: GST053 and MIFTS: 87.

#### 4.1.2. Common Genes Analysis

The comparison between QRC target genes and the genes related to GC was made using the Venny 2.1 website. Here, both sets of genes gathered in QRC Targed Genes and GC Target Genes were manually added to it to generate a Venn diagram.

#### 4.1.3. Gene Ontology and Functional Annotation Assay

Using genes common to GC and QRC, the ShinyGo 0.77 [46], and the DAVID-Bioinformatic Resources [47] websites were used for the gene ontology and Functional Annotation Assay. For ShinnyGo 0.8, the parameters were set to Homo Sapiens species with a false discovery rate (FDR) threshold of 0.05 percent, and to calculate the fold enrichment (FE) of each target with a false discovery rate (FDR) [48] threshold of 0.05. Results with an FE equal to or greater than 5 were used to ascertain the biological pathways implicated by those genes using the Kyoto Encyclopedia of Genes and Genomes (KEGG) database. For the Functional Annotation Assay, the DAVID-Bioinformatic Resources website was used; for this, we added the common genes between GC and QRC to the website, obtaining Functional Annotations results.

#### 4.1.4. Protein-to-Protein Analysis

To obtain HUB genes from the combination of GC and QRC, a protein-to-protein (PPI) network was used. The StringDB website [49,50,51,52,53,54,55,56,57] was employed by adding all the genes previously analyzed in the gene ontology assay and then from the proteins obtained in this analysis, those with at least 10 interactions between them were designated as HUB genes.

#### 4.1.5. Survival Curves and Immune System Infiltration

The GEPIA2 database [58] was used to obtain survival curves of HUB genes in GC, with a significance level of *p* ≤ 0.05.

The TISIDB [29] database was used for immune system infiltration, with *p* ≤ 0.05 as significant, and using the Spearman test for correlation.

#### 4.1.6. Molecular Docking Analysis

Using the SwissDock website (from the Swiss Institute of Bioinformatics) [44], we analyzed the genes that were associated with cancer invasion and metastasis by filtering the HUB genes with the software Cytoscape (Versión 3.10.1) (https://cytoscape.org/ accessed 18 March 2024) [59]. This required pdb files of HUB gene proteins, obtained from the Research Collaboratory for Structural Bioinformatics [60] and AlphaFold Protein Structure Database [61]. As for the QRC structure, a mol2 file was obtained from PubChem [62] and converted to the required format using the Department of Internal Medicine Translational Informatic Division’s converting tool (https://datascience.unm.edu/tomcat/biocomp/convert) (accessed 18 March 2024). Results were visualized using UCSF Chimera software using UCSF CHIMERA software Version 1.17.3 (https://www.cgl.ucsf.edu/chimera/docs/ContributedSoftware/webdata/webdata.html#browser (accessed 4 April 2024) [63]. The results shown are only those with a ΔG ≤ −6.

## 5. Conclusions

According to all the results of this study, QRC has the potential to be at least a co-adjuvant factor in the treatment of GC. It has a wide spectrum of potential targets that have been associated with multiple types of cancer survival, proliferation, and evolution of this disease. This study also provides a more generalized view of the effects of the compound on other cells that also directly or indirectly affect cancer, such as immune system cells. This is particularly interesting since, for instance, while most of the HUB genes are not linked to improved survival of GC (with the notable exception of AXL), the number of immune cells that can change its infiltration is outstanding and this reinforces the idea that QRC could have its place as a complement to a pharmacological treatment more than previously thought. This study as a whole also gives an idea of the bioinformatic studies of natural biocompounds in cancer since it provides a guide of how to take not only a particular focalized affected cell or system against a compound but also takes notice of other important parts of the disease (like immune system infiltration) and analyzes them in conjunction without obtaining so much data that they become unrealistic to integrate into the findings.

Finally, it is important to remark that, even with all the results presented in this paper, at the end of the day, by its nature, it has some limitations. First, while bioinformatic tools are incredibly useful in obtaining a more precise idea of complex problems, such as the effects of a molecule in a disease (QRC and GC), to make further research more focused, it is important to remark that the information is still incomplete. For instance, in this article, more than 30 possible biological targets of QRC were detected, but to date, there is little to no information about most of them in GC. Therefore, while this study accomplishes its general objective, it is only a step toward the bigger picture of effectively using QRC in the treatment of GC.

## Figures and Tables

**Figure 1 ijms-25-07934-f001:**
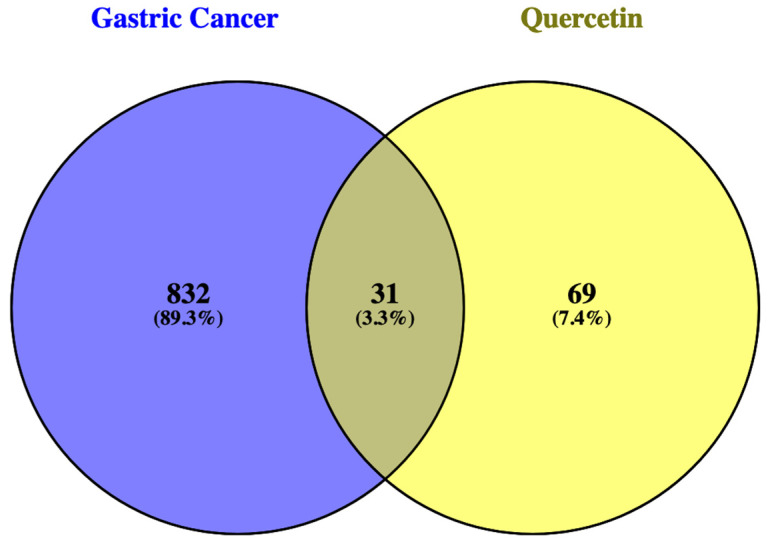
Genes in common between GC and QRC using the Venny 2.1. website. This figure compares the total number of genes gathered from Swiss Target Prediction (for QRC) and Malacards (for GC) and shows the overlap between them in the intersection.

**Figure 2 ijms-25-07934-f002:**
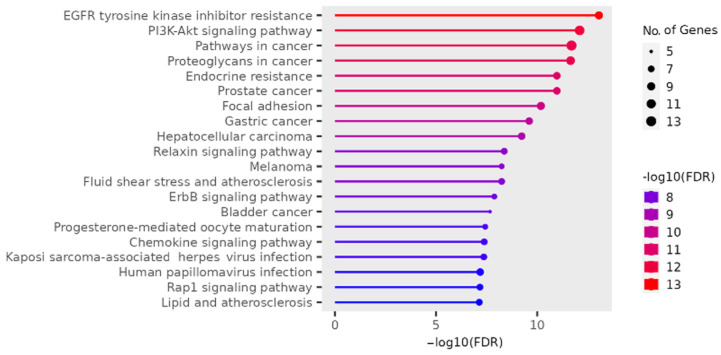
Pathway Assessment Chart of the 31 genes in common between QRC and GC sorted by FDR according to the Shinny Go 0.8 website. The further to the right the pathway, the higher its FDR; this is also highlighted by the color of the graph, and the size of the circle at the point of the graph indicates how many of the tested genes are involved in that pathway.

**Figure 3 ijms-25-07934-f003:**
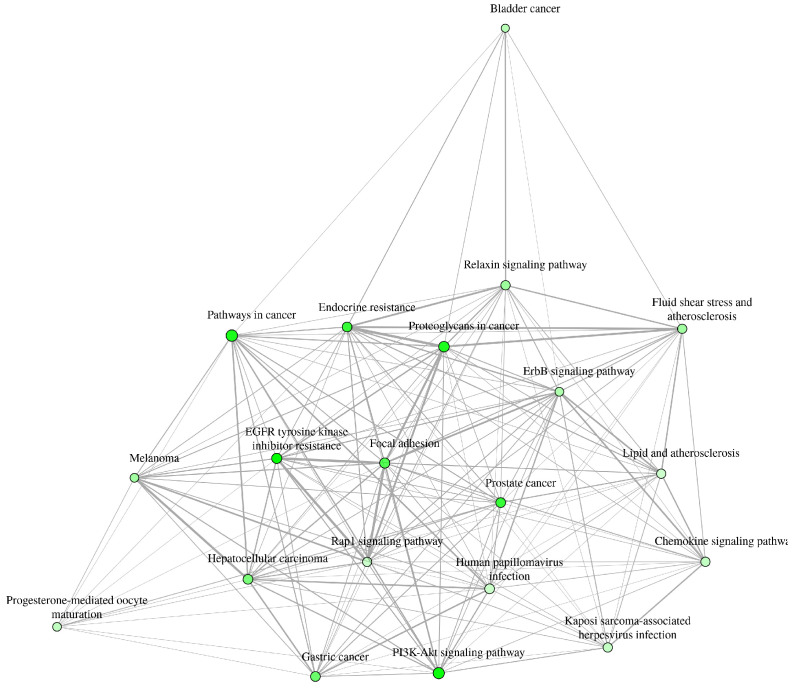
Pathway Network Analysis of genes in common between GC and QRC. The relationships between each pathway suggested by the Shinny Go 0.8 website are linked by the lines. Furthermore, the intensity of the color in each pathway is linked to its FDR (the higher the FDR, the stronger the color).

**Figure 4 ijms-25-07934-f004:**
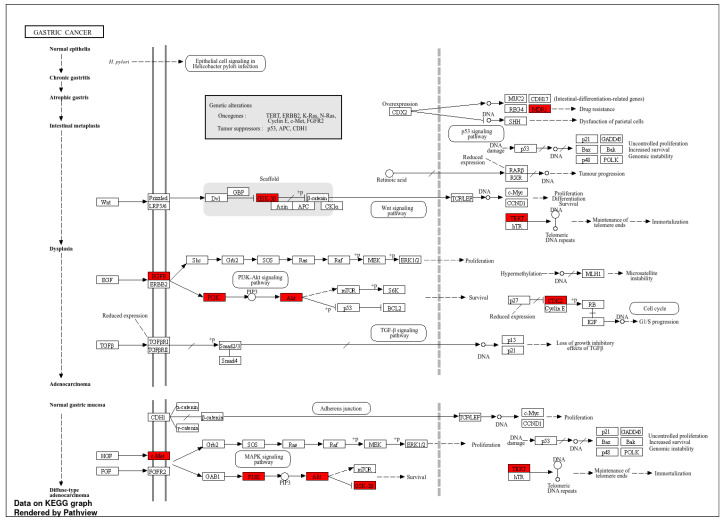
GC pathway with possible QRC target genes highlighted in red according to both the KEGG and Shinny Go 0.8.

**Figure 5 ijms-25-07934-f005:**
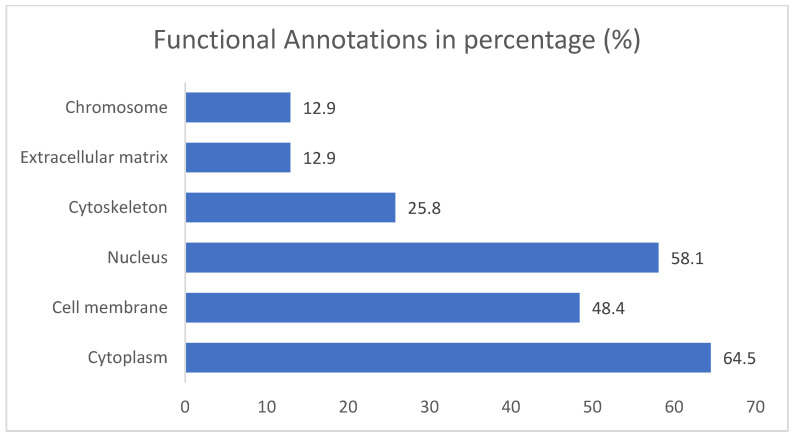
Functional annotations of cellular components related to genes in common between GC and QRC according to the DAVID Bioinformatics database.

**Figure 6 ijms-25-07934-f006:**
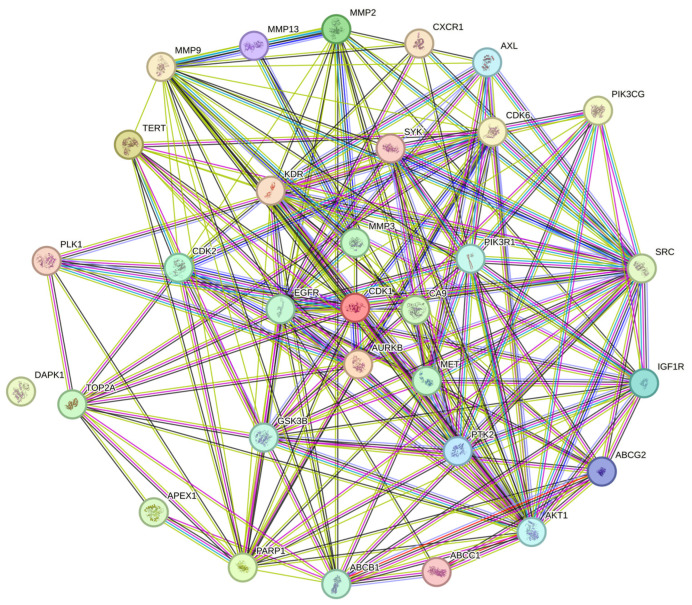
The protein-to-protein analysis of GC and QRC target genes. Total nodes: 31, edges: 192, average node degree: 12.4, avg. local clustering coefficient: 0.734, expected edges: 74, PPI enrichment *p*-value: <1.0 × 10^−16^. As about the colors of the lines, those indicate interactions sourced from well bpupulated databases (blue); predicted interactions within each neighborhood gene (green), gene fusion, gene concurrence (red, and navy-blue respectively). Additional text-mining, co-expression, and protein homology, (borders in grass green, black, and gray represent respectively).

**Figure 7 ijms-25-07934-f007:**
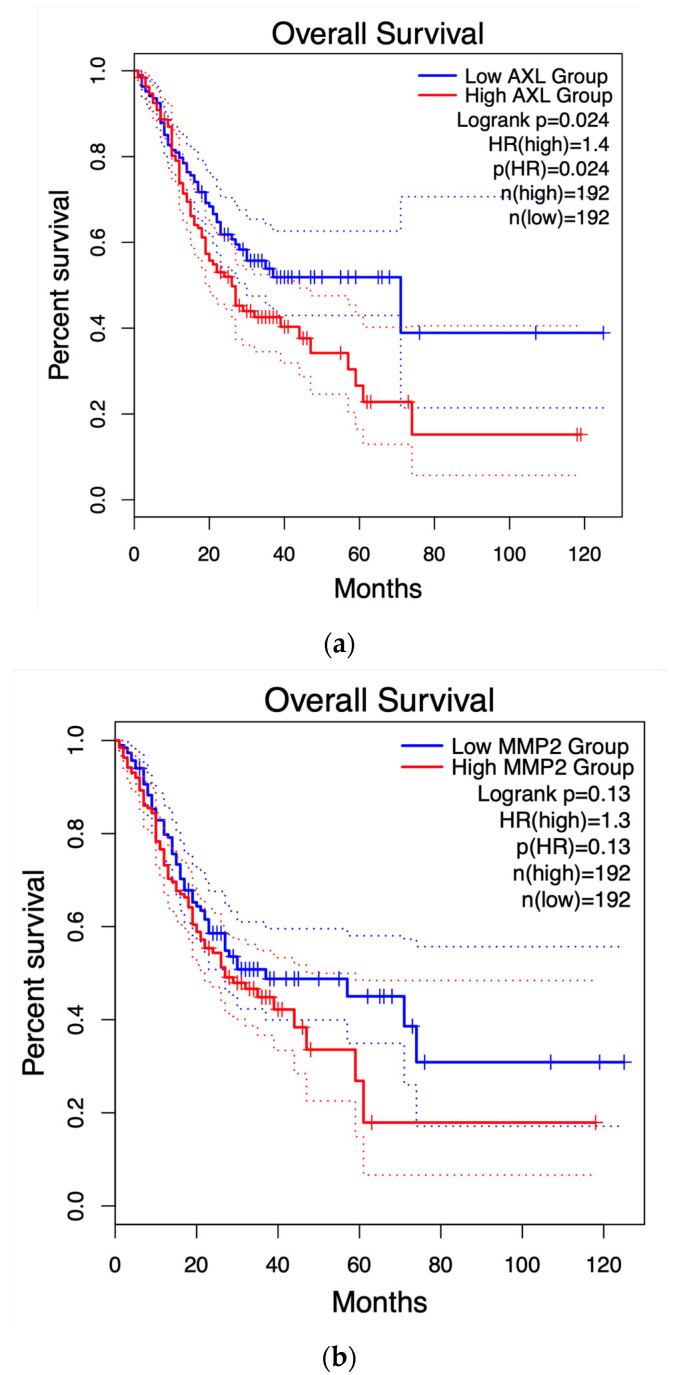
Survival curves of the HUB genes according to the GEPIA 2 database; this analysis, (**a**) shows the results for AXL, the only HUB gene with a *p* < 0.05, (**b**) shows the results of MMP2, (**c**) shows the results for CDK2.

**Figure 8 ijms-25-07934-f008:**
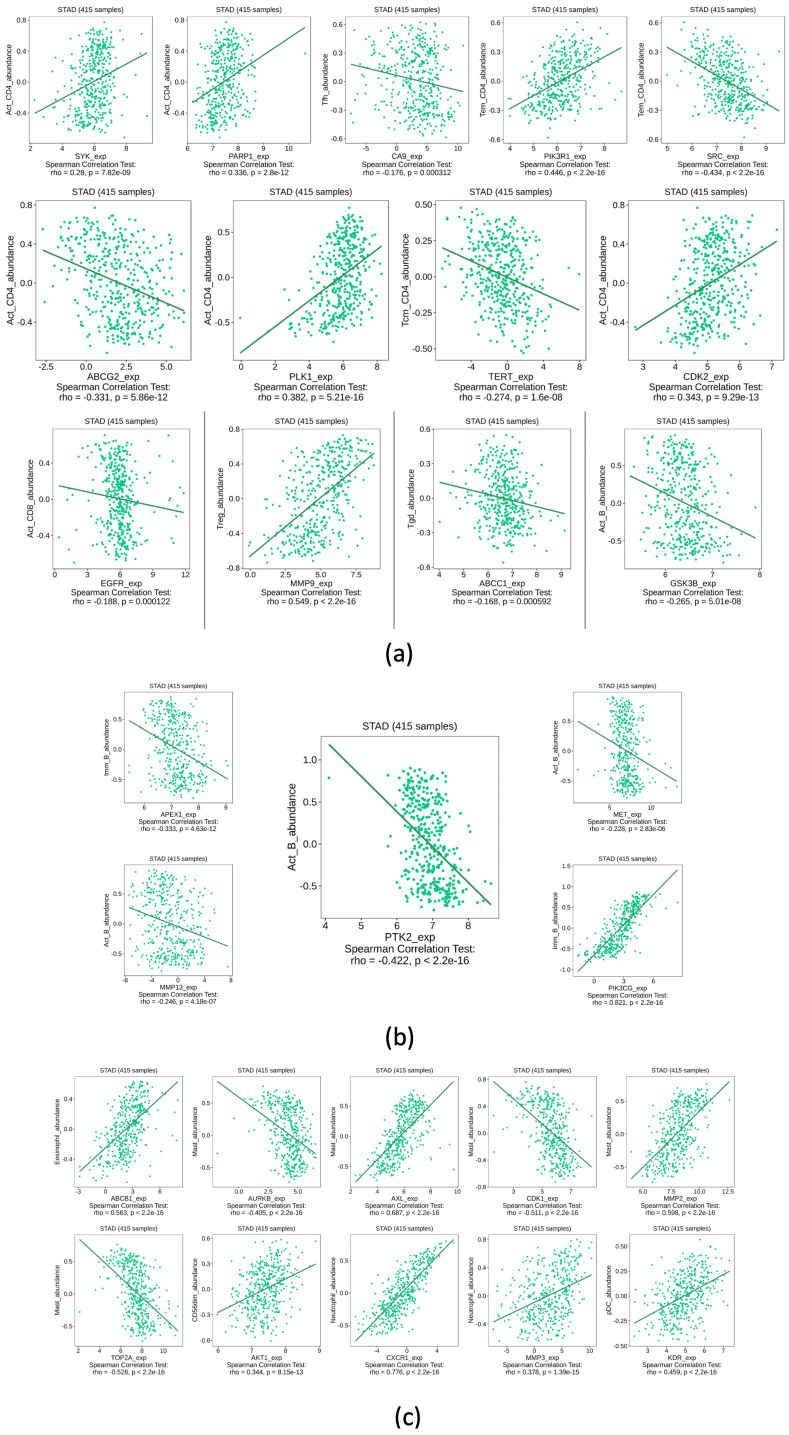
Full immune system infiltration analysis segmented by cell type. All genes showed a *p* ≤ 0.05 and their highest or lowest rho score from −1 to 1. (**a**) T-cell-related genes. (**b**) B-cell-related genes. (**c**) Other immune system cell-related genes.

**Figure 9 ijms-25-07934-f009:**
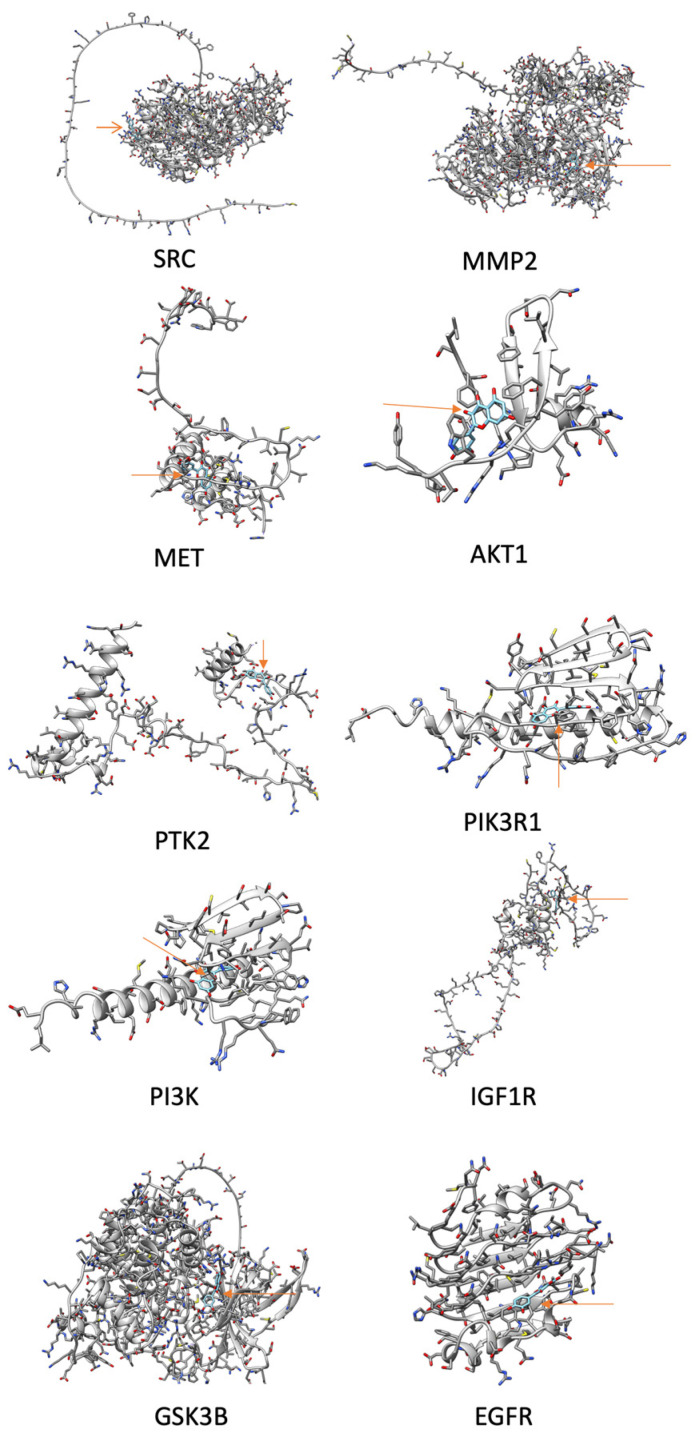
Docking images obtained from UCSF Chimera using information from SwissDock. The protein genes are shown mostly in gray while QRC is light blue with an orange arrow pointing to it. Only the molecules with a ΔG ≤ −6 are shown.

**Figure 10 ijms-25-07934-f010:**
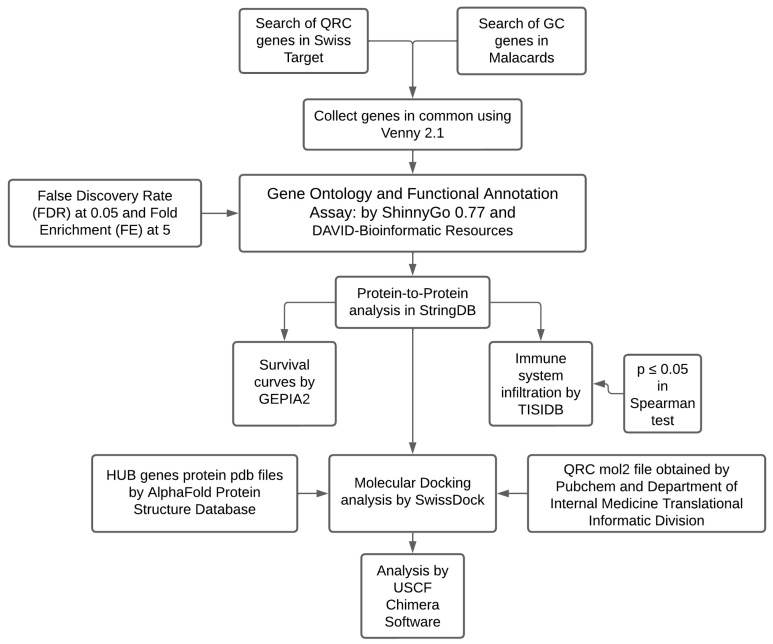
Complete design of the study.

## Data Availability

The original contributions presented in the study are included in the article; further inquiries can be directed to the corresponding authors.

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
