# Peer review of "A Bioinformatic Assay of Quercetin in Gastric Cancer"

_ijms, 2024, doi:10.3390/ijms25147934_

Round 1

Reviewer 1 Report

Comments and Suggestions for Authors

In the manuscript submitted for review, the Authors analyzed a bioinformatics assay of quercetin in gastric cancer.

 I find the topic of the manuscript interesting and "up-to-date", and the entire work is well-thought-out. The Authors put a lot of work into preparing this interesting work. The carefully prepared engravings undoubtedly attract the reader's attention.

My comments:

1. in my opinion, the main thing missing from the introduction is information on how quercetin has anti-cancer properties. I think such information is necessary.

2. figure 2 - in my opinion, requires a detailed description

3. line 138 - part of the text is covered by the chart

4. most figures require a detailed legend

Author Response

  1. Commentary: in my opinion, the main thing missing from the introduction is information on how quercetin has anti-cancer properties. I think such information is necessary.
    Response: Thank you for pointing this out. We agree with the suggestion so, we have increased the information regarding QRC and cancer, you can find it in lines 51 to 55.
  2. Commentary: figure 2 - in my opinion, requires a detailed description
     Response: Thanks again for pointing this out, we have made some changes, first of all, figures have been re-ordered so, what was figure 2 now is figure 1. Even so, we made a more detailed description. This is found in line 75. 
  3. Commentary: line 138 - part of the text is covered by the chart
    Response: Thank you for pointing this out. We have reorganized all the information on the paper, so there should be no issue with this or anything similar
  4.  Commentary: most figures require a detailed legend
    Response: Thank you for this comment, we have made changes to most figures so they have a. more detailed legend. 

Reviewer 2 Report

Comments and Suggestions for Authors

Comments on the manuscript ijms-3097812, this is an interesting in silico analysis where the authors evaluate the therapeutic potential of QRC in the treatment of GC using bioinformatics tools to analyze the impact of QRC on GC cells, examining its effects on the cellular pathways and gene expression. Some comments are listed below:

Scientifically validate the websites used in the study including the address, of websites and  date of access

The paragraphs that include the material and methods require improving their writing and explanation at each stage of the analysis.

Figure 3 is confusing especially in the N. of genes part and the black circles are not marked in the figure

Figure 4 does not exist

Improve the resolution of figure 5, it is overlapped with part of the text

Figure 7 analyzes the 31 protein-to-protein target genes for quercertin and gastric cancer. However, the figure is not clear. It is suggested that the names be added outside the figure for better visualization and greater explanation.

Figure 8 requires a better description to understand the importance of survival, the description of the protein groups AXL, MMP2, CDK2 is missing, although the annexes try to explain it, it is not understood due to the low quality of the figure.

Figure 10 in the text refers to 13 hub genes and only shows 10 to explain the absence of the missing ones

Discussion. The first paragraphs of the discussion are not precise (251-269) it describes results instead of enriching its discussion

Much of the discussion refers to figure 4, which is not visible, which detracts from its understanding.

To understand the discussion of its results, it is suggested to put it by subtopics, not by figures.

Improve your conclusion according to the computational results obtained and mention the limitations after the discussion. Include comments on how this analysis can improve a non-bioinformatics essay.

Author Response

1.- Comment: Scientifically validate the websites used in the study including the address, of the websites and  date of access
Response: This has been addressed, in the section on results there are the web links to all the pages and their results are also cited as the software owners required. 

2.- Comment: The paragraphs that include the material and methods require improving their writing and explanation at each stage of the analysis.
Response: Thank you for pointing this out. We have changed and improved the method segment. 

3. Comment: Figure 3 is confusing, especially in the N. of genes part and the black circles are not marked in the figure
Response: Thank you for pointing this out, we have reorganized all the figures so now what was Figure 3 is now Figure 2. Also, we have improved the description of the figure. 

4.- Comment: Figure 4 does not exist
Response: Thank you for pointing this. We realized there were many issues with the figures showing, so all of them have been corrected, what was figure 4 now is figure 3 and it should be correct. 

5. Comment: Improve the resolution of figure 5, it is overlapped with part of the text
Response: This has been noticed and fixed

6. Comment: Figure 7 analyzes the 31 protein-to-protein target genes for quercetin and gastric cancer. However, the figure is not clear. It is suggested that the names be added outside the figure for better visualization and greater explanation.
Response: Thanks for noticing this, we have made some modifications so the names are now legible and not cropped. Also, the figure now is Figure 6. 

7. Comment: Figure 8 requires a better description to understand the importance of survival, the description of the protein groups AXL, MMP2, CDK2 is missing, although the annexes try to explain it, it is not understood due to the low quality of the figure.
Response: Thank you for this comment. Most figure descriptions have been improved and the quality of the images should have been improved where needed, also, this figure has been re-numbered to Figure 7.  

8. Comment: Figure 10 in the text refers to 13 hub genes and only shows 10 to explain the absence of the missing ones
Response: Thanks for pointing this out, we had pointed out in methods that this figure would only show those with a delta G equal or higher than 6, we made sure to point this out better so it is understood. 

9. Comment: Discussion. The first paragraphs of the discussion are not precise (251-269) it describes results instead of enriching its discussion
Response: Thank you for this comment. We have taken notice and see your point. However, we consider that the beginning of the Discussion is adequate for a better understanding of the idea of this work as a whole. 

10. Comment: Much of the discussion refers to figure 4, which is not visible, which detracts from its understanding.
Response: Thank you for pointing this out. We have made changes needed to fix this (figure 4 is now figure 3). 

11. Comment: To understand the discussion of its results, it is suggested to put it by subtopics, not by figures.
Response: Thanks for your suggestion, we took and have made changes in the discussion so most paragraphs are introduced for their subtopic and not their corresponding figures. 

12. Comment: Improve your conclusion according to the computational results obtained and mention the limitations after the discussion. Include comments on how this analysis can improve a non-bioinformatics essay.
Response: Thank you for this comment. We have made some changes in the conclusion so it better encompasses the whole paper and, it also gives a suggestion of how our work can improve bio-informatic jobs.